# Toward Standardization of a Lung New Approach Model for Toxicity Testing of Nanomaterials

**DOI:** 10.3390/nano14231888

**Published:** 2024-11-24

**Authors:** Elisabeth Elje, Laura M. A. Camassa, Sergey Shaposhnikov, Kristine Haugen Anmarkrud, Øivind Skare, Asbjørn M. Nilsen, Shan Zienolddiny-Narui, Elise Rundén-Pran

**Affiliations:** 1Norwegian Institute for Air Research, 2027 Kjeller, Norway; eel@nilu.no; 2National Institute of Occupational Health in Norway, 0033 Oslo, Norway; laura.camassa@stami.no (L.M.A.C.);; 3NorGenoTech, 0349 Oslo, Norway; 4Faculty of Medicine and Health Sciences, Department of Clinical and Molecular Medicine, Norwegian University of Science and Technology, 7491 Trondheim, Norway

**Keywords:** ALI, NAMs, genotoxicity, NM-300K, A549, triculture

## Abstract

This study represents an attempt toward the standardization of pulmonary NAMs and the development of a novel approach for toxicity testing of nanomaterials. Laboratory comparisons are challenging yet essential for identifying existing limitations and proposing potential solutions. Lung cells cultivated and exposed at the air-liquid interface (ALI) more accurately represent the physiology of human lungs and pulmonary exposure scenarios than submerged cell and exposure models. A triculture cell model system was used, consisting of human A549 lung epithelial cells and differentiated THP-1 macrophages on the apical side, with EA.hy926 endothelial cells on the basolateral side. The cells were exposed to silver nanoparticles NM-300K for 24 h. The model used here showed to be applicable for assessing the hazards of nanomaterials and chemicals, albeit with some limitations. Cellular viability was measured using the alamarBlue assay, DNA damage was assessed with the enzyme-modified comet assay, and the expression of 40 genes related to cell viability, inflammation, and DNA damage response was evaluated through RT^2^ gene expression profiling. Despite harmonized protocols used in the two independent laboratories, however, some methodological challenges could affect the results, including sensitivity and reproducibility of the model.

## 1. Introduction

New approach methodologies (NAMs) are needed for the hazard identification and characterization of chemicals for next generation risk assessment (NGRA). NAMs include in silico, in chemico, and in vitro assays, and high-throughput screening and omics approaches [1]. An important part of NAMs is the development of advanced in vitro models, like 3D models consisting of several cell types, better representing tissue and organ structure. Standardization and validation of such advanced in vitro models are of high importance and in line with the 3R’s principle (replacement, reduction and refinement of methods which avoid the use of animals) [2].

Inhalation is the most important human exposure route for airborne particular matter (PM), including nanomaterials (NMs) [3,4]. In vitro models for use in inhalation toxicology include cell lines or primary cells cultured in 2D and 3D, or cocultures with static or dynamic medium flow. Human cell lines, such as A549, BEAS-2B, and Calu-3, are commonly used in models for respiratory toxicology. Increased human pulmonary relevance of the models is achieved when culturing the cells on permeable inserts at the air-liquid interface (ALI), due to better physiological conditions with tighter cell-to-cell connections [5,6]. To further enhance the physiological relevance and predictivity of the model, the respiratory tract cells can be cultured in combination with other cell types, like endothelial and immune cells [6]. ALI culture models are also beneficial for particle exposure, as the physicochemical properties of NMs are likely to be less affected when exposure in the cell culture medium is avoided [6].

New methodologies require validation and standardization prior to implementation in risk assessment, and interlaboratory studies are needed to evaluate the usefulness, robustness, and reliability of the models [2,6]. The most frequently studied endpoints in interlaboratory comparisons of the NM effects are inflammatory markers and cytotoxicity [7,8,9]. A recent interlaboratory comparison study with seven participating laboratories, studying a mono- and coculture model of Calu-3 cells with and without macrophages, showed promising results for the transferability of the model and use in cytotoxicity [8]. A study with two participating laboratories evaluated the interlaboratory variability of submerged and ALI cultured A549 cells and highlighted the need for detailed protocols and training of personnel [9]. There is a need for more knowledge on interlaboratory comparison studies on advanced ALI models to establish validated standard operating procedures (SOPs), especially in relation to the application of genotoxicity test methods.

This study aimed to investigate the properties of the triculture lung model through a comparative standardization study conducted across two laboratories. For this work, we used silver nanoparticles measuring less than 20 nm, with NM-300K from the JRC nanomaterial repository as a reference particle substance. NM-300K has previously been extensively used in our laboratories with the model [10,11]). We optimized the SOP to enhance robustness and harmonize the results.

Using a PBS buffer diluted 1:10 in milli-Q water as internal control at the ALI, we successfully created a nebula in the VITROCELL^®^ cloud chamber. There was no significant reduction in cell viability compared to the unexposed control in both laboratories. Despite following the same protocol for the particle suspension and dosing of NM-300K, discrepancies in cell viability and DNA damage assay outcomes were observed across the laboratories. To gain further insight, we also performed a medium-throughput gene expression profiling to evaluate changes in the expression of key genes related to cell viability, inflammation, and genotoxicity. In the apical A549/dTHP-1 cells, exposure to NM-300K led to alterations in inflammatory biomarkers. In the basolateral EA.hy926 cells, we measured an upregulation in the expression of the inflammatory CXCL1 and oxidative stress HMOX1 genes.

Our findings emphasize the importance of further standardizing and validating advanced in vitro models, like the ALI triculture system, to ensure their reliability for the safety evaluation of chemicals, including NMs in future implementation in NGRA as an alternative to animal testing.

## 2. Materials and Methods

### 2.1. Cell Cultures

The human alveolar type II lung epithelial A549 cells [12], endothelial EA.hy926 cells [13], and monocytic THP-1 cells [14] were purchased from the American Type Culture Collection (ATCC, Manassas, VA, USA). The cell lines were cultured in Dulbecco’s modified Eagle’s medium (DMEM) or Roswell Park Memorial Institute (RPMI), supplemented with 9–10% fetal bovine serum (FBS, product no. 26140079, ThermoFisher Scientific, Oslo, Norway; FBS, ultra-low endotoxin product no: S009Y20008, Biowest, VWR, Oslo, Norway) and 1% penicillin/streptomycin (Pen-Strep, product no. 15070063, ThermoFisher Scientific), and maintained in an incubator with a humidified atmosphere at 5% CO_2_ and at 37 °C. The same batch of cells were used for the experiments performed at both laboratories (Laboratory 1: STAMI, Laboratory 2: NILU). For more information see Camassa and Elje et al., 2022 [10].

The preparation of tricultures on permeable Falcon 6-well inserts (1 µm pores) was performed as described [10]. In this study, we used a seeding density of 1.1 × 10^5^ cells/cm^2^ A549 cells and 2.2 × 10^5^ cells/cm^2^ differentiated THP-1 cells on the apical side, and 1.1 × 10^5^ cells/cm^2^ EA.hy926 cells on the basolateral side of the inserts. An identical differentiation protocol for converting THP-1 cells from monocytes to macrophage-like cells (dTHP-1) was employed in both laboratories. For the differentiation process, phorbol-12-myristate-13-acetate (PMA, Sigma-Aldrich, St. Louis, MO, USA, product no. P8139, EU) was prepared as a stock solution at a concentration of 1 mg/mL in dimethyl sulfoxide (DMSO). This stock solution was then diluted in Milli-Q (MQ) water to achieve a final concentration of 10 μg/mL, and aliquots were stored at −20 °C in the dark. PMA was added to undifferentiated THP-1 cells at a concentration of 50 ng/mL for a duration of 3 days. The dTHP-1 cells were subsequently cultured for 48 h in RPMI complete medium before harvesting [10].

Harvesting of differentiated THP-1 cells before seeding onto membranes was performed by using Accutase (Sigma A6964) only, without cell scraping as used in the previous study [10]. We initially employed a ratio of 2 dTHP-1 cells to 1 A549 cell (when at confluency), as some cells were lost during medium removal or damaged during the detachment procedure prior to seeding. We observed that the actual ratio of dTHP-1 to A549 was significantly lower and more closely aligned with the in vivo conditions [10] (Appendix A).

In tricultures, the cells were cultured in a coculture medium consisting of 72% DMEM high glucose (Gibco 11965-092, Thermo Fisher Scientific, Oslo, NO), 18% DMEM low glucose (Gibco 31885023, Thermo Fisher Scientific, Oslo, NO), 10% heat-inactivated FBS, and 1% pen-strep. A549, EA.hy926, and THP-1 were used at passages (P) 6–15, 6–15, and 6–21, respectively (details in Appendix A). All cell lines, before use, were tested regularly for mycoplasma contamination and found to be negative.

### 2.2. Immunofluorescence

The immunofluorescence procedure has been previously described in Camassa and Elje et al., 2022 [10]. At 24 h postexposure, unexposed and ALI exposed tricultures were washed in PBS and fixed with a solution of 4% paraformaldehyde (PFA) in PBS for 15 min at RT. The membranes were removed from the plastic holder with a blade and kept in a 1:10 solution of fixative and PBS. The cells were permeabilized with a solution of 0.01% Triton-X 100 in PBS for 10 min. Unspecific epitopes were saturated in 2% BSA in PBS for 30 min. Cells were incubated overnight in a humidified chamber with primary antibodies directed toward cell markers specific to the air-blood barrier (Appendix A). The primary antibodies were diluted in 2% BSA in PBS. The membranes were washed in PBS and incubated with secondary antibodies conjugated to a fluorescent probe (Appendix A). Cells were stained with the nuclear staining DAPI for 5 min, washed in PBS and mounted on a microscopy slide with a mounting medium solution (Invitrogen, Waltham, MA, USA) and covered with a coverslip. The cells were examined with a confocal Zeiss L-10 inverted microscope (Appendix A).

### 2.3. Nanomaterials

The JRC Repository NM, NM-300K, was provided from Fraunhofer IME (Schmallenberg, Germany). NM-300K, as previously described in Camassa and Elje et al., 2022 [10], consists of engineered spherical silver nanoparticles with a pristine size of less than 20 nm and was selected as a reference nanoparticle due to its demonstrated toxicity in previous studies [10,15,16,17]. NM-300K is suspended in a colloidal dispersion medium composed of 85% deionized water, 7% stabilizing agent (ammonium nitrate), and 8% emulsifiers (4% Polyoxyethylene Glycerol Trioleate and 4% Polyoxyethylene Sorbitan Monolaurate, Tween 20). NM-300K has a nominal silver concentration of 10% (*w*/*w*) (Joint Research et al., 2011). The same batch of NM-300K was utilized by both laboratories.

The NANOGENOTOX protocol was employed for the dispersion of NM-300K. A solution of 6 mL with a final concentration of 10 mg/mL was prepared in water containing 0.05% bovine serum albumin (BSA) through sonication, as detailed in Camassa and Elje et al., 2022 [10]. Following sonication, the NM-300K dispersion solution was diluted in a 1:10 PBS solution using MQ water. Consequently, we further minimized the concentration of BSA in the solution that we nebulized.

The dispersed NMs were stored on ice and vigorously vortexed before physicochemical characterization and use for cell exposures shortly after preparation.

The hydrodynamic diameter and size distribution of the NM stock dispersion was measured by dynamic light scattering (DLS). The dispersion was diluted in ultrapure water (1:100) by pipetting, transferred to a disposable cuvette (DTS0012), and placed in the Zetasizer (Zetasizer Ultra Red or Nano ZS, both from Malvern Panalytical Ltd., Malvern, UK). Measurements were performed as previously explained [10]. Results are presented as Z-average (Z-ave), which is the intensity weighted mean hydrodynamic size of the ensemble collection of particles, and polydispersity index (PDI).

Zeta potential (ZP) was measured at the same dilution as for size analysis. The diluted dispersion was transferred to a disposable folded capillary cell (DTS1070) pre-wetted with ethanol and water and placed in the Zetasizer. The ZP was measured by mixed-mode measurement–phase analysis light scattering (M3-PALS) at 25 °C with 3–5 parallel measurements and 120 s equilibration time.

### 2.4. Exposure at the Air-Liquid Interface

For exposure of the cells at the ALI, the commercially available VITROCELL^®^ 6 Cloud System (VITROCELL^®^ Systems GmbH, Waldkirch, Germany) with an Aerogen Pro^®^ (VITROCELL^®^ Systems GmbH, Waldkirch, Germany) vibrating membrane nebulizer was used. The system was maintained at 37 °C in a laminar flow hood. Transwell inserts with tricultures were transferred to the VITROCELL^®^ 6 Cloud System, which was filled with 18 mL of cell culture medium/well to let the basolateral side of the insert be in contact with medium.

The triculture models were exposed at the ALI to aerosols of NM-300K (diluted to 7 mg/mL in sterile milli-Q H_2_O) as a reference material with a nominal concentration of 20 µg/cm^2^, or to PBS 10% (diluted 1:10 in sterile milli-Q H_2_O) as a negative control. For calculations on nominal exposure concentrations, see the Appendix A. The deposition of the sample was controlled in semi-real time by a quartz crystal microbalance (QCM). Before nebulization, the sample was vortexed for 15 s.

The exposure was performed by nebulizing a total volume of 800 µL (4 × 200 µL with 1 min pause between each) of exposure solution in the chamber, and letting the cloud settle for 5 min before opening the chamber. The inserts were transferred to 6-well plates with fresh culture medium (1.5 mL, basolateral side only). Unexposed control cultures (incubator control) were also transferred to a new plate with fresh culture medium. The cultures were placed in the incubator for 20–24 h before processing for further analysis. The QCM measurement was stopped after a stable signal 5–7 min after opening the exposure chamber.

Cell exposures were performed in the same order in each experiment: first, PBS control (10% in H_2_O), then NM-300K. Between each experiment, the nebulizers were rinsed with milli-Q water for several minutes and sonicated in a water bath followed by an ethanol bath (70%) for 10 min.

### 2.5. AlamarBlue Assay

After 20–24 h of exposure, the cellular viability was measured by the metabolic activity assay alamarBlue on both the apical and basolateral sides of the triculture inserts. The cultures were washed with PBS before being incubated with 1–1.5 mL of alamarBlue solution (10%, *v*/*v*) for 1–1.5 h followed by fluorescence reading and analysis. Positive controls for cytotoxicity were included in both laboratories before incubation with alamarBlue. In Laboratory 1, separate lung triculture inserts were treated with a lysate solution (10×) (ThermoFisher) for 45 min. In Laboratory 2, separate triculture inserts were exposed for 20–24 h to 100 µM chlorpromazine hydrochloride (cat. no. C8138, Sigma-Aldrich) in the basolateral culture medium. For further details on the procedure and interference control, see Camassa and Elje et al., 2022 [10].

### 2.6. Harvesting of Cells from the Lung Triculture Model

Immediately after performing the alamarBlue assay, the lung cell cultures were gently washed with PBS (1 mL on the apical side, 2 mL on the basolateral side). The cells were wet trypsinized for 5–10 min in the incubator, with 500 µL trypsin (0.25%, Sigma-Aldrich, St. Louis, MO, USA) on the apical side and 1 mL trypsin (0.05%, Sigma) on the basolateral side of the membrane. The trypsin was neutralized by 2 mL of medium in each compartment. The cells were resuspended by gentle pipetting and transferred to separate Eppendorf tubes. The cells were centrifuged at 300× *g* for 10 min before the pellet was resuspended in 2 mL of PBS. The total number of cells and the cell viability was measured by trypan blue staining in Countess 3 (Invitrogen, Waltham, MA, USA). The cell suspensions were further used for the comet assay (Section 2.6) in both laboratories and for RNA extraction (Section 2.7) in Laboratory 2. In Laboratory 1, 24 h after exposure, lung cell cultures were gentle washed with cold PBS (1 mL on apical side and 2 mL on basolateral side), and cells from the apical and basolateral sides of the inserts were scraped with lysis buffer (Qiagen, Hilden, Germany) and kept at −80 °C until RNA extraction.

### 2.7. Comet Assay

The miniaturized 12-gel enzyme-modified version of the comet assay was performed to measure DNA damage (strand breaks, SBs) and oxidized base lesions in the ALI-cultured cells, as described previously [10,18]. The cells from tricultures were harvested as explained above and stored on ice for a maximum of 3 h. The comet assay was performed by NorGenoTech for samples from Laboratory 1 and by NILU for samples from Laboratory 2, by application of the same SOP [10,18,19]. The cells were centrifuged, resuspended, and diluted in PBS to approximately 200.000 cells/mL. Aliquots of the cell suspension were mixed with low melting point agarose (0.8% *w*/*v*, Sigma-Aldrich, 37 °C) to a final agarose concentration of 0.64% *w*/*v*. Mini-gels (10 µL) with approximately 400 cells were made on cooled microscopic slides pre-coated with 0.5% standard melting point agarose (Sigma-Aldrich), with a maximum of 12 gels per slide. Slides were placed in Coplin jars and submerged in lysis solution (2.5 M NaCl, 0.1 M EDTA, 10 mM Tris, 1% *v*/*v* Triton X-100, pH 10, 4 °C) for a maximum of three days. As a positive control for DNA strand breaks (SBs), separate slides were submerged in 100 µM H_2_O_2_ (in PBS, 4 °C) for 5 min, rinsed twice with PBS, and then submerged in a separate Coplin jar with lysis solution.

The modified comet assay was used with the bacterial repair enzyme formamidopyrimidine (Fpg) DNA glycosylase (NorGenoTech, Oslo, Norway) for detection of oxidized or alkylated bases. Fpg converts oxidized or alkylated purine bases to SBs [20]. Further steps of the comet assay were performed as described previously, including details on incubation with Fpg, DNA unwinding (20 min), electrophoresis (1.25 V/cm, 20 min), and neutralization in PBS and H_2_O [10,18]. Internal control cell samples were included for positive control of Fpg (cells with and without Ro 19-8022 and light treatment), as described previously [10]. Comets were scored using the software Comet assay IV 4.3.1 (Perceptive Instruments, Bury St Edmunds, UK). Median DNA tail intensity, proportional to the number of SBs, was calculated from 50 comets per gel as a measure of DNA SBs. Medians were averaged from 2 to 4 gels per cell culture insert. A total of 3–4 independent experiments were performed with single or duplicate inserts per experiment. Control for possible interference between NM-300K and analysis of comets has been tested previously and no interference was found [10].

### 2.8. Gene Expression Profiling RT^2^

RT^2^ is a highly sensitive and reliable method to investigate gene expression. RT^2^ profile PCR arrays combine the RT-PCR performance with the ability to detect the expression of many genes simultaneously. A customized plate of forty genes was investigated in two exposure sets in two laboratories. The forty genes analyzed can be divided into five pathway groups of biomarkers involved in inflammation, fibrosis, cell death, cellular oxidative stress, and DNA damage.

The profile RT^2^ array was run on cell lysates from the apical and basal side of the 3D lung model after 24 h post-exposure in ALI of 1:10 PBS, NM-300K at 20 µg/cm^2^ and unexposed control (NC). The cell lysates of the apical and basal side of the 3D lung model were compartmentalized in the analysis.

At 24 h post-exposure, triculture samples were washed twice in PBS and cells were lysed with 350 µL of buffer RLT (Qiagen AllPrep RNA/DNA kit, Qiagen, Hilden, Germany) with 20 mM of dithiothreitol (DTT, Sigma-Aldrich) from the apical compartment (A549 /dTHP1) and the basolateral compartment (EA.hy926) (Section 2.5). The cell lysates were frozen at –80 °C before further analysis. Total RNA from the cell lysates was isolated with Qiagen AllPrep RNA/DNA kit (Qiagen).

The nanodrop ND1000 spectrophotometer (Thermo Fisher Scientific, Oslo, NO, USA) was used to determine the purity of the extracted RNA. The concentration of RNA was measured using the Qbit 4-Fluorometer (Thermo Fisher Scientific) following manufacture’s guidelines.

Overall, 400 ng of total RNA was used for a 384 plate and was reverse transcribed with RT^2^ first strand kit cDNA synthesis (Qiagen). The kit includes a buffer to eliminate any potential genomic DNA contamination that would cause false positive signals. An ulterior reverse-transcription control (RTC) is present inside each of the 384 plates. Samples that result positive for the genomic contaminations were excluded from the gene expression analysis. A customized setup for the 48 genes in the 384-well plate was provided. Each set of 48 genes contained 5 reference genes and 3 internal control genes: RTC, GDC, a genomic DNA control specific for the species, hGDC, and a PCR positive control (PPC), which consists of a DNA sequence detected by the polymerase.

The 5 reference genes used in the analysis are peptidylprolyl isomerase A (PPIA), phosphoglycerate kinase 1 (PGK1), hydroxymethylbilane synthase (HMBS), transferrin receptor (TFRC), and peptidylprolyl isomerase A (PPIA). The analysis was made into an RT^2^ Profiler PCR Array Data Analysis Web portal (Qiagen) to upload files exported from the QuantStudio 5 Real-Time PCR System (Applied Biosystems, Foster City, CA, USA), Thermo Fisher Scientific). Gene expression analysis was done on the cycle threshold (Ct) values, defined as the cycle where a statistically significant increase in fluorescence above the background signal is detected. The Ct values are inversely proportional to the concentration of the genes. The geometric mean of the reference genes expression (Ct of the reference genes) was used to normalize the expression of the other genes in all the groups (ΔCt). Ct values ≥ 33.5 were set as undetermined and not considered in the analysis. ΔΔCt are raw data and are defined as the ratio of the relative gene expression between the control group and the exposed group. Changes in genes expression were analyzed in fold change (2^−ΔΔCt^). Numbers equal to or greater than 2-fold indicate a biological upregulation or increase in gene expression, and numbers equal to or lower than 0.5-fold indicate a biological downregulation or decrease in gene expression, and a fold change of ±1 indicates no change in gene expression [21].

### 2.9. Statistics

A total of n = 3–5 independent experiments were performed in each laboratory, each with 1–2 culture inserts per treatment group. In Laboratory 1, n = 5 for alamarBlue assay, and n = 4 for comet assay SBs, n = 4 for comet assay SBs + Fpg, and n = 3 for gene expression. For each condition, 2 culture inserts were used. In Laboratory 2, n = 4 for alamarBlue assay, comet assay and gene expression. For each condition, 2 culture inserts were used, except in two experiments for NM-300K where 1 culture insert was used.

AlamarBlue and comet assay: Results are presented as mean with a standard deviation (SD) of n = 3–5 independent experiments. Statistical analysis of alamarBlue and comet assay results was performed by comparing the mean of each sample to the mean of negative control (inserts exposed to PBS 10% in milli-Q water) by one-way ANOVA with multiple comparisons and post-test Tukey using GraphPad Prism version 9.3.1 for Windows, GraphPad Software, San Diego, CA, USA. The statistical analysis of comet assay results with reference cells with and without Ro 19-8022 treatment was performed using one-way ANOVA with multiple comparisons and post-test Tukey. The level of significance was set to *p* < 0.05.

Gene expression profiling RT^2^: We calculated the fold change (2^−ΔΔCt^) and their corresponding *p*-values for each gene using a linear mixed model of the log-fold values (ΔCt). This model included groups as a fixed effect, enabling a comparison of each exposed group with a control group. Additionally, a random intercept was included for each experiment. The significance level was defined as *p* < 0.05. Each group consisted of n = 6 samples. For some genes that were up- or downregulated based on the criteria, a Student’s *t*-test was subsequently conducted between the exposed group NM-300K and the PBS-exposed group. The number of independent experiments per group was n = 3, with 2 replicates per group. The Student’s *t*-test was performed on the mean values of fold change (2^−ΔΔCt^) with a 95% confidence interval (CI) and two-tailed *p*-values.

## 3. Results

### 3.1. Nanomaterial Characterization

The NM-300K was characterized in both laboratories using the same protocol. The physicochemical properties are shown in Table 1, indicating a broad size distribution and a semi-stable dispersion. The particle size distribution of most dispersions had multiple peaks, indicating the presence of some aggregated particles.

### 3.2. Cellular Viability and DNA Damage

Exposing the cells at ALI to the PBS negative control (PBS 10% in Milli-Que (MQ) water) induced an expected small reduction in the relative cell viability, compared to the incubator unexposed control (NC), in both laboratories (±20%) (Appendix A). In Laboratory 1, this reduction was significant (Appendix A). In Laboratory 1 and 2, respectively, the viability was 77.6% ± 20.8% and 78.3% ± 10.1% for apical cells, and 76.7% ± 7.4% and 75.8% ± 20.4% for the endothelial cells (Figure 1). Positive controls (PC) induced a significant reduction in the relative cell viability in both laboratories (Appendix A and Figure 1A,B). In Laboratory 1, exposure to 20 µg/cm^2^ NM-300K reduced cell viability at the apical (A549/THP1) and basolateral side (EA.hy926) compared to PBS controls, but not in Laboratory 2 (Appendix A and Figure 1A,B).

NM-300K induced a significant increase in DNA strand breaks (SBs) and on oxidized DNA lesions in EA.hy926 cells, while no effect was observed in A549/dTHP-1 cells (Appendix A and Figure 2B). In Laboratory 1, no significant effect was seen in DNA strand breaks (SBs) or on oxidized DNA lesions when exposed to NM-300K in both apical or basolateral sides (Appendix A and Figure 2A). The positive controls to test the activity of the Fpg enzyme were Ro 19-8022 exposed cells (TK6 cells in Laboratory 1 and A549 cells in Laboratory 2 showed induction of strand breaks as expected based on historical control data in the laboratories) (Appendix A and Figure 2). The positive control H_2_O_2_ resulted in a high level of SBs (>80%) in accordance with our historical control data.

### 3.3. Gene Expression Profiling RT^2^

A customized panel of forty genes was investigated using medium-throughput gene expression profiling RT^2^. The forty genes analyzed can be divided into five groups of biomarkers involved in inflammation, fibrosis, cell death, cellular oxidative stress, and DNA damage.

The gene expression profiling RT^2^ array was performed on RNA samples isolated from cell lysates at the apical and basolateral sides of the lung triculture cell model after 24 h NM-300K post-exposure. The fold changes in gene expression were compared to the PBS exposure control group apical or basolateral sides (Figure 3 and Figure 4). Genes with a more than 0.5-fold decrease in expression were classified as downregulated, while those with a more than 2-fold increase in expression were classified as upregulated.

For Laboratory 1, Figure 3A showed a heatmap indicating the expression profiling RT^2^ of all forty genes at the apical and basolateral sides of the lung triculture cell model. Statistical analysis was performed on genes where the expression at the apical and basolateral sides showed either up- or a downregulation compared to their expression in the PBS-exposure control groups (Figure 3B). At 24 h NM-300K post-exposure, in Laboratory 1, CXCL1 gene expression showed an upregulation of 3.78-fold at the apical side and 3.99-fold on the basolateral side. At the apical side, the ATM gene expression was upregulated by 2.43-fold. At the basolateral side, the NF-κΒ gene was upregulated by 2.74-fold, the HMOX1 gene was upregulated by 16.17-fold, and the PRKDC gene was upregulated by 2.28-fold compared to their expression in the PBS control (Figure 3B).

Gene expression profiling using RT^2^ array for Laboratory 2 is depicted in the heatmap in Figure 4A. Figure 4B displays the statistics of the genes which expression, at the apical and basolateral sides of the lung triculture model, was up- or downregulated compared to the PBS controls (Figure 4B).

Downregulated gene expression was found at the apical side for the inflammation marker genes IL1-β by 0.27-fold, IP-10 by 0.22-fold, and RANTES by 0.45-fold, the fibrosis marker gene cd11b1 by 0.3-fold and the cell death marker gene BCL-2 by 0.43-fold (Figure 4B). At the basolateral side, the inflammation marker gene CXCL1 was upregulated by 2.36-fold), the oxidative stress marker gene HMOX1 by 10.9-fold, and the cell death marker genes RPK1 by 1.8-fold and Caspase 9 by 1.6-fold (Figure 4B).

Summaries of the results for the gene expression changes, cytotoxicity (alamarBlue), and DNA damage (comet assay) for Laboratories 1 and 2 are shown in Table 2 and Table 3, respectively.

#### Gene Expression Comparisons Between Controls in Laboratory 1 and in Laboratory 2

We compared the fold change of gene expression (2^−ΔΔCt^) in NCs (unexposed) and PBS exposure groups between the laboratories (Figure 5).

NC and PBS exposure from Laboratory 1, and PBS exposure from Laboratory 2 were compared to NC from Laboratory 2 to assess if the two test analyses were comparable to each other.

The test set at the apical side, with NC from Laboratory 1 and PBS exposures from Laboratory 1 and 2, showed significant differences in gene expression for most of the biomarkers when compared with NC from Laboratory 2 (Figure 5A). A downregulation in the inflammatory biomarkers was measured to NC from Laboratory 1 and PBS exposure from Laboratory 2 when compared with NC from Laboratory 2 (Figure 5A). Upregulation was seen in biomarkers related to DNA damage, such as Chk1 and BRCA2 in NC from Laboratory 1 and after PBS exposures in Laboratory 1 and 2, in comparison with NC from Laboratory 2 (Figure 5A).

The test set at the basal side, with NC from Laboratory 1 and PBS exposure from Laboratory 1 and 2, showed significant differences compared to the NC from Laboratory 2 (Figure 5B).

A downregulation in VEGF, RANTES, MIP 1β, CXCL1, and PDGFA was seen only for NC and PBS exposure from Laboratory 1; a downregulation in SP-D was observed in NC from Laboratory 1 and both PBS exposures from Laboratories 1 and 2; upregulation in SP-A was observed in NC and PBS exposure from Laboratory 1 when compared to NC of Laboratory 2 (Figure 5B).

NCs apical and basal from Laboratory 2, biomarkers gene expression, were significantly different when compared to NC and PBS exposed from Laboratory 1. The data on the apical and basal side were therefore difficult to compare between laboratories (Figure 5).

## 4. Discussion

Development and standardization of advanced in vitro models are critically important for the implementation of NAMs in NGRA. Alveolar cells cultured at the ALI have previously been shown to be a promising model for the hazard assessment of chemicals [2,5,6,7,8,9]. In this study, we investigated the performance of an alveolar triculture model of epithelial A549, endothelial EA.hy926, and macrophage-like (differentiated) THP-1 cells prepared in two laboratories using similar procedures. We have compared results focused on viability, DNA damage, and gene expression profiles in negative controls and after aerosol exposure to a reference NM-300K silver nanoparticle.

We previously compared three alveolar ALI models with different layers of complexities after exposure to NM-300K, with a focus on cyto- and genotoxicity [10]. Commonly used A549 cells were cultured in monoculture and in coculture with endothelial EA.hy926 on the basolateral side of the membrane, or in triculture with the addition of macrophage-like differentiated THP-1 cells on the apical side. We found that the effect of NM-300K exposure changed by adding complexity by introducing several cell types. The cellular viability after NM-300K exposure was significantly reduced in monocultures only, and an increase in DNA damage was found only in the endothelial cells of cocultures, although the particles were shown to be internalized in the endothelial cells in the triculture model [10].

Based on our previous study [10], we have further optimized the cultivation and exposure protocols. Difficulties with harvesting a representative amount of the differentiated THP-1 cells to be seeded in the tricultures were avoided in the current study by using a fresh and room-tempered Accutase solution, without the need of cell scraping. This allowed all experiments to have the same cell numbers, which is important for harmonizing the results and reducing the variability in the responses. We also used very similar passage numbers of cells for the experiments in the two laboratories.

In the exposure protocol, the exposure volume and nebulization time were extended to increase the concentration of NM-300K from 10 µg/cm^2^ to 20 µg/cm^2^. Consequently, we have used the same volume and time for PBS exposure. However, in the present study, we have diluted the PBS to 10% in sterile MQ-H_2_O. This measure ensures that the cells were not exposed to excessively high salt concentrations while still securing a proper cloud formation for the exposure control sample.

Exposure to PBS induced a slight reduction in cellular viability compared to the incubator control. This effect was significant only for Laboratory 1. However, both laboratories had harmonized results on the cellular viability measurements in the PBS control, which is an improvement from the previous study [10]. For the positive control of the assay, we have used different approaches in the laboratories, with lysis of cells in Laboratory 1 and treatment to a chemical in Laboratory 2. Both approaches successfully reduced the viability to a significantly low number, which shows the proper function of the alamarBlue assay.

To gain a deeper understanding of the model performances in the two laboratories, we conducted gene expression profiling RT^2^. We examined forty genes that can be divided into five interconnected groups of biomarkers related to inflammation, fibrosis, cell death, cellular oxidative stress, and DNA damage. We compared gene expression of the five pathways between negative controls from Laboratory 1 and 2, and we observed a significant difference for some of the inflammatory biomarkers on the apical A549 and dTHP-1 cells (Figure 5). The use of differentiated THP-1 cells can introduce additional challenges in the reproducibility of results between laboratories [22], as they have shown varying responses to both NMs and positive controls [22,23].

THP-1 cells belong to a group of macrophages and immune cells that are sensitive to endotoxin levels, which may necessitate serum with low endotoxin [24]. Fetal bovine serum (FBS) provides essential nutrients and growth factors for cell maintenance and growth. Commercially available standard FBS can have a concentration of ≤10 EU/mL, corresponding to approximately 1 ng/mL of endotoxin [24]. Xia et al., 2013 emphasized the importance of using the same batch of serum and cells for consistent interlaboratory results when working with differentiated THP-1 cells [7]. In our study, we used the same batch of cells, and medium with the same final composition, but from different batches. However, the serum was not identical in both laboratories. The serum was heat-inactivated when used in Laboratory 1, as heating inactivates complement proteins. Active complement proteins can participate in inflammatory events and activate lymphocytic and macrophage cells [25].

To test the toxic responses in the triculture lung models, the cells were exposed to aerosols of NM-300K. The effect of NM-300K on cellular viability and induction of DNA damage differed in the two laboratories. A reduction in viability was observed in both cell compartments in Laboratory 1, while an increase in DNA SBs and oxidized base lesions was detected in the basolateral EA.hy926 cells in Laboratory 2 only. In the comet assay, it is crucial to use non-cytotoxic concentrations to prevent false positive results [26]. The lack of induction of SBs in Laboratory 1 could be due to the measured cytotoxicity, which was not seen in Laboratory 2. Previous experience with NM-300K in 2D cultures has identified a steep toxicity curve/narrow interval for the start of the cytotoxic effect [17,18,19], which may contribute to high variability in the results. Potential interference between NM-300K and the alamarBlue or comet assays has been previously investigated and no interference was found [10], and is thus not likely to explain the observed differences in response. However, the detected basal differences in gene expression levels could influence the response of the cells to the treatments (Figure 5 and Table 2 and Table 3).

Further, our results emphasize the importance of physicochemical characterization of the dispersed particles when performing toxicity testing. The hydrodynamic diameter (Z-ave) of the nanosilver particles was found to be smaller in Laboratory 1 than in Laboratory 2, even though we used the same dispersion protocol with harmonized calibration of the sonicators and similar DLS instruments and measurement techniques. The size distributions indicated the presence of larger particles, suggesting some aggregation. The difference in the size of the NMs could influence the toxicity [27].

The cellular uptake of NMs is important to include in genotoxicity studies for the accurate interpretation of results [28,29,30]. Internalization of NM-300K was confirmed in the apical cells in our previous study, where particles appeared to be localized inside lamellar body vesicles as single particles (5–20 nm) or in small aggregates of about 100 nm. No particles were found in the endothelial cells [10].

To further investigate the reasons for the differences in toxic response measured, we performed gene expression analysis for comparison of the profile in the exposure control and exposed cells in both laboratories. Interestingly, differences in the gene expression profiling were found between the laboratories. On the apical side, we observed variation in the fold change of inflammatory markers at 24 h after exposure to NM-300K. An upregulation in the chemokine CXCL1 and ATM was observed in Laboratory 1 and in Laboratory 2, and we observed a downregulation in inflammatory, fibrosis and cell death biomarkers IL-1β, IP-10, RANTES, Cd11b1, and BCL-2 when compared to its own exposure control.

On the basolateral side, where the endothelial cells exist in an undisturbed environment, the effects are influenced by perturbations on the apical side. In both Laboratory 1 and Laboratory 2, we observed an upregulation in the chemokine CXCL1, involved in inflammation and in fibrosis, and the HMOX1 biomarker, involved in DNA damage and oxidative stress pathways, at 24 h post-exposure. In Laboratory 1, we noted an upregulation in NF-κB and PRKDC, while in Laboratory 2, there was an upregulation in RPK1 and Caspase 9.

Previous studies demonstrated an upregulation in oxidative stress under exposure to AgNPs [31]. Oxidative stress is a cellular state in which the amount of reactive oxygen species (ROS), caused by NM´s oxidative potential can lead to the oxidation of essential biomolecules, causing inflammation and oxidative DNA damage. Oxidative stress is an important key event in the mode of action of many NMs, such as silver nanoparticles [32,33], and serves as an early warning indicator.

The differences between laboratories were already observed in previous studies that investigated inflammatory biomarkers in intra- and interlaboratory comparisons [7,8,9,23]. These differences could be explained by various factors, such as the use of different serum [34], THP-1 cell density and attachment [8], environmental humidity [35], or deposited doses in the ALI system [36].

A comparison between laboratories is challenging. Several limitations were identified in our study, such as differences in serum, which could explain the increased gene expression of inflammatory biomarkers in the controls from Laboratory 2 compared to those from Laboratory 1 at the apical side, where dTHP-1 cells were seeded alongside A549 cells (Figure 5). A limiting factor in maintaining the density of dTHP1 in the triculture lung model is the adherence of these cells. In the study by Camassa and Elje et al. (2022) [10], despite using a cell density of 2 dTHP1 to 1 A549 cell, we observed that after air lifting, the number of dTHP1 cells adhering to and remaining in the culture was estimated to be lower than the actual concentration of dTHP1 that had been seeded. This evaluation was conducted through microscopic analysis, comparing the observed numbers to the initial seeding counts [10]. A similar challenge was reported in the paper by Braakhuis et al. (2023), where dTHP1 cells were cocultured with Calu-3 cells [8].

The production of surfactant by A549 cells at the apical side may also influence the results, sensitivity, and reproducibility of the model, particularly in the air-liquid interface (ALI) system. In vivo, alveolar type II epithelial cells synthesize and secrete surfactant proteins in the alveolar space. Although the A549 cell line has been shown to produce surfactant in previous studies, varying cell culture conditions may challenge surfactant production. This could result in an unbalanced and insufficiently moist cellular environment, which is essential for a three-dimensional (3D) lung alveolar in vitro model. Even so, we did not observe any dryness at the apical side of our model.

Previously, we demonstrated through electron microscopy and dark-field microscopy that in our 3D lung alveolar model, both dTHP-1 and A549 cells engulf NM-300K silver nanoparticles [10]. In vivo, the majority of nanoparticles engulfed in the alveolar space are taken up by alveolar macrophages, with a lesser extent occurring through transcytosis by the alveolar epithelium [37]. The differences in cellular uptake of silver nanoparticles between different cell types and among laboratories may also account for the variations in the data obtained within the model.

The protocol limitations identified across laboratories should be addressed in future validation and development of the 3D lung alveolar model. Nevertheless, we believe that this study is important for enhancing knowledge and developing new methodological approaches.

## 5. Conclusions

The development and validation of new approach methodologies (NAMs) are crucial for their use in next generation risk assessment (NGRA) of nanomaterials and other chemicals. This study aimed to standardize an alveolar triculture lung model across two laboratories, focusing on the responses of epithelial A549, endothelial EA.hy926, and macrophage-like THP-1 cells to NM-300K silver nanoparticles exposure at the air-liquid interface (ALI). The goal was to improve reproducibility and reliability of the lung triculture model for in vitro hazard assessment.

We sought to standardize culturing and exposure procedures between laboratories to ensure harmonized results. This study evaluated cytotoxicity and genotoxicity as key endpoints, while also using the RT^2^ profiling PCR array to analyze several toxicological pathways from the same samples. This array was applied for laboratory comparison to enhance predictive accuracy. Additionally, the internal control in the ALI system was standardized by reducing PBS concentration to 10%, preventing high salt exposure to the cells and enabling meaningful comparisons of cell viability against an unexposed incubator control.

Our results highlight the critical role of standardized protocols in complex models to achieve reproducible outcomes. We recognized the importance of using the same supplement type, particularly for sensitive cell lines as THP-1. Optimized protocols, including consistent cell numbers and exposure parameters, reduced variability compared to previous studies. However, differences in gene expression, especially in inflammation markers, persisted between laboratories, likely influenced by factors such as serum, cell number, and nanoparticle characteristics. These findings emphasize the importance of robust models and protocols for reproducibility and the need for consistent physicochemical characterization and model optimization to ensure reliable hazard assessments in NAMs.

Our data demonstrate that the lung triculture model is a promising NAM for assessing human hazard from inhalation exposure to chemicals and nanomaterials. This aligns with the 3R’s principle, promoting the transition from animal-based experimental testing to the use of NAMs. This study marks an important step toward the standardization of pulmonary NAMs for assessing the risks of nanomaterial exposure.

## Figures and Tables

**Figure 1 nanomaterials-14-01888-f001:**
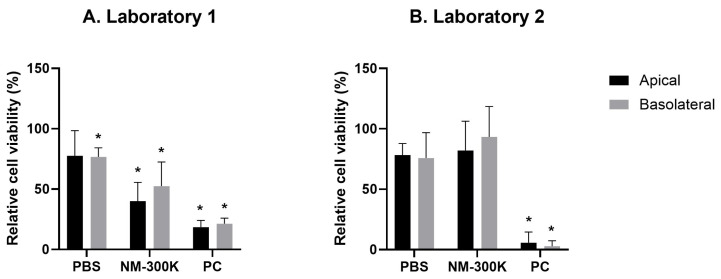
Relative cell viability (%) toward NC (set at 100%) measured by the alamarBlue assay at the apical and basolateral sides of the lung triculture cell model in Laboratory 1 (**A**) and Laboratory 2 (**B**). The analysis was conducted 24 h post-exposure with NM-300K (20 µg/cm^2^) and PBS negative control (PBS 10% in MQ water) at the air-liquid interface (ALI). PC = positive control, which for Laboratory 1 was Triton X-100, and for Laboratory 2 was chlorpromazine hydrochloride at 100 µM. The columns represent the mean values and the bars standard deviation from a total of n = 5 (**A**) and n = 4 (**B**) independent experiments with n = 2 replica culture inserts in each experiment. Statistics conducted by one-way ANOVA with post-test Tukey with a significance level of * *p* < 0.05. Additional data are presented in Appendix A.

**Figure 2 nanomaterials-14-01888-f002:**
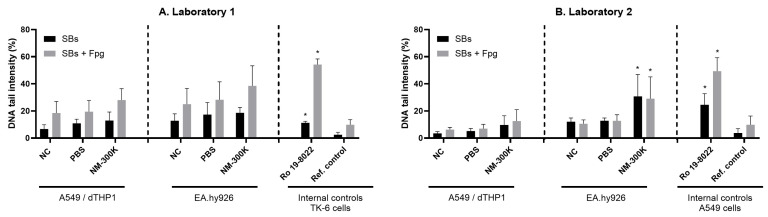
DNA strand breaks (SBs) (black) and oxidized DNA lesions (SBs + Fpg) (gray) determined by the enzyme-modified comet assay on apical (A549/dTHP-1) and basolateral (EA.hy926) cells from the lung triculture model at the air-liquid interface as % of DNA in tail, in two laboratories (**A**,**B**). The columns represent the mean values, and the bars show standard deviation from a total of n = 4 independent experiments with n = 2 replica inserts for each experiment. Statistics were conducted using one-way ANOVA with post-test Tukey with a significance level of * *p* < 0.05. Additional numbers and statistics are presented in Appendix A.

**Figure 3 nanomaterials-14-01888-f003:**
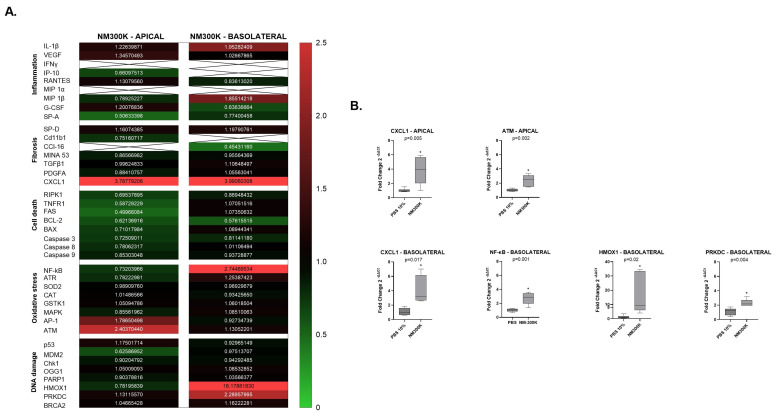
Gene expression RT^2^ array analysis of different groups of genes involved in inflammation, fibrosis, cell death, oxidative stress, and DNA damage. The data shown are from Laboratory 1. The number of independent experiments is n = 3, with 2 replicates per group. (**A**) Mean values in the heatmap correspond to fold change (2^−ΔΔCt^) in the expression of 40 genes in the NM-300K 20 µg/cm^2^ exposed groups, both apical (A549/THP-1 cells) and basolateral (EA.hy926 cells), compared to their respective expression in the PBS control group, for Laboratory 1. The heatmap is based on a double gradient: the largest value on the scale 2.5, on the left of the heatmap in vertical, is indicated in red, the baseline value of 1 in black, and smallest value of 0 in green. Values larger than 2 are in bright red and are upregulated genes, while values smaller than 0.5 shows downregulated genes in green. Ct-Threshold (cycle threshold) = 33.5; x-box = low or not expressed. (**B**) Among the 40 genes expression in fold change (2^−ΔΔCt^) of CXCL1 and ATM were significantly upregulated at the apical side after exposure to NM-300K. At the basolateral side, significant upregulated gene expression in fold change (2^−ΔΔCt^) was measured for CXCL1, NF-κB, HMOX1 and PRKDC Statistical analysis (GraphPad): student’s *t*-test of mean values of fold change (2^−ΔΔCt^) with a 95% confidence interval (CI) and two-tailed *p*-values indicated in each graph. (*) indicates significance. Boxes and whisker plots respectively indicate variance and minimum to maximum values, with the line in the box representing the mean of the values.

**Figure 4 nanomaterials-14-01888-f004:**
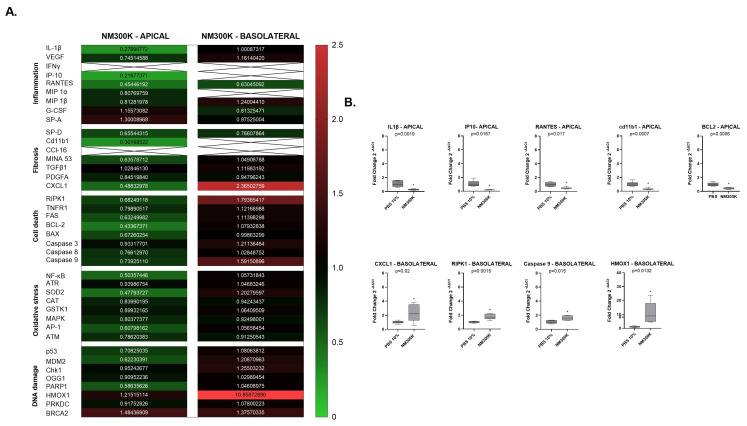
(**A**) Gene expression RT^2^ array analysis of different groups of genes involved in inflammation, fibrosis, cell death, oxidative stress and DNA damage. Data shown are from Laboratory 2. The number of independent experiments is n = 3; with 2 replicates per group. (**A**) Mean values in the heatmap correspond to fold change (2^−ΔΔCt^) in the expression of 40 genes in the NM-300K 20 µg/cm^2^ exposed groups, both apical (A549/THP-1 cells) and basolateral (EA.hy926 cells), compared to their respective expression in the PBS control group for Laboratory 2. The heatmap is based on a double gradient: the largest value on the scale is 2.5, on the left of the heatmap in vertical, indicated in red, with the baseline value of 1 in black, and the smallest value of 0 in green. Values higher than 2 are in bright red and are upregulated, while values lower than 0.5 are downregulated and in green. Ct-Threshold = 33.5; x-box = low or not expressed. (**B**) On the apical side, gene expression of IL 1β, IP10, RANTES, Cd11b1, and BCL-2 were significantly downregulated. On the basolateral side, gene expression of CXCL1, RIPK1, Caspase 9, and HMOX1 were upregulated. Statistical analysis (GraphPad): student’s *t*-test of mean values of fold change (2^−ΔΔCt^) with a 95% confidence interval (CI) and two-tailed *p*-values indicated in each graph. (*) indicates significance. Boxes and whisker plots indicate variance and minimum to maximum values, respectively; the line in the box represents the mean of the values.

**Figure 5 nanomaterials-14-01888-f005:**
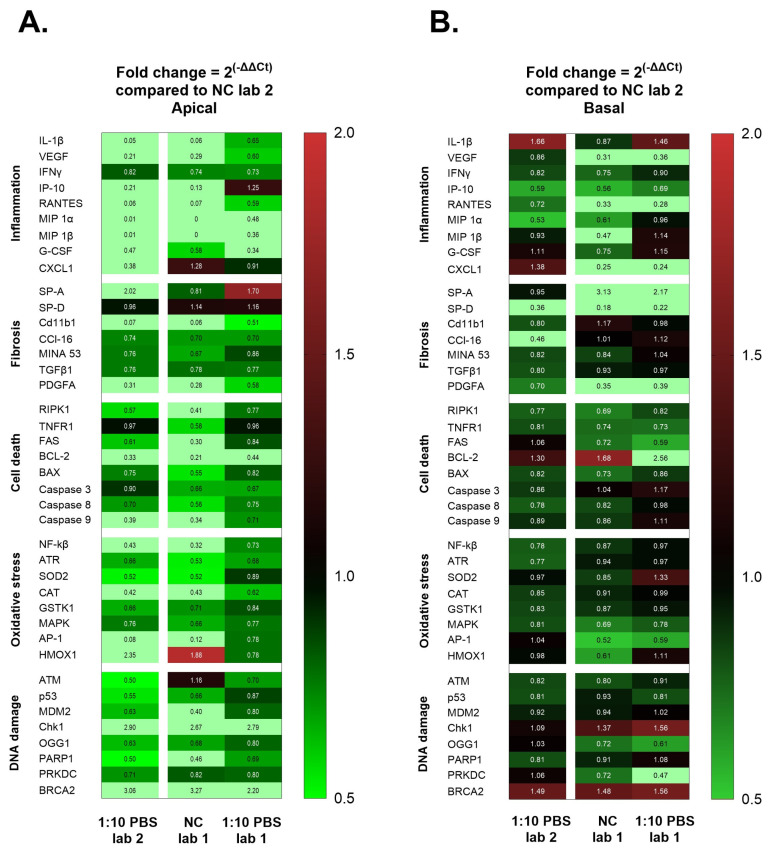
Heatmap of fold change (2^−ΔΔCt^) gene expression analysis compared to unexposed control (NC) Laboratory 2. (**A**) Apical cells (A549/dTHP-1) and (**B**) basal cells (EA.hy926). 1:10 PBS Lab 2: 1:10 PBS exposure Laboratory 2; NC Lab 1: NC Laboratory 1; 1:10 PBS Lab 1: 1:10 PBS exposure Laboratory 1. The heatmap is based on a double gradient: the largest value on the scale is 2 on the left of the heatmap in vertical, indicated in red, the baseline value of 1 is in black, and the smallest value of 0.5 is in green. Values larger or smaller out of the scale are in bright green, and when higher than 2 are upregulated, or smaller than 0.5 are downregulated. Fold change (2^−ΔΔCt^) statistical analysis was made using gene globe expression profiling analysis provided by Qiagen. Student’s *t*-test on the replicate 2^−ΔΔCt^ values for each of the 40 genes in each of the exposed group (NC) compared to the control group in Laboratory 2. N of experiment per group = 3; 2 replicates per group.

**Table 1 nanomaterials-14-01888-t001:** Nanomaterial physicochemical characterization data of NM-300K stock dispersion (10 mg Ag/mL in 0.05% BSA water, diluted to 7 mg/mL in sterile Milli-Que water) measured by DLS. Results are presented as mean with standard deviation of the defined number of experiments (n). BSA: bovine serum albumin; DLS: dynamic light scattering; ZP: zeta potential in millivolt (mV); *w*/*w*: weight/weight; PDI: polydispersity index in arbitrary units (a.u.); Z-ave: average of aerodynamic size in nanometer.

NM-300K ^1^, Physicochemical Characterizations by DLS Before Exposure		
Parameters	Laboratory 1	Laboratory 2
Z-ave (nm)	71.9 ± 2.9 (n = 4)	107.4 ± 47.1 (n = 3)
PDI (a.u.)	0.372 ± 0.068 (n = 4)	0.396 ± 0.078 (n = 3)
ZP (mV)	−16.1 ± 0.9 (n = 3)	−18.6 ± 0.4 (n = 3)

^1^ NM-300 silver nanoparticles from JRC. Stability, homogeneity; JRC scientific and technical report [11].

**Table 2 nanomaterials-14-01888-t002:** Summary of results for Laboratory 1 (Lab 1). Changes in gene expression (RT^2^ profiling assay) and in cytotoxicity (alamarBlue), and DNA damage (comet assay), 24 h after air-liquid interface exposure with PBS at 10% and 20 µg/cm^2^ of NM-300K. Exposure groups are compared to NC from Laboratory 1.

RT^2^ Profiling Assay Lab 1				
NM-300K Exposure Toward PBS 10% Lab 1				
Regulation	Apical		Basolateral	
	Down-	Up-	Down-	Up-
Inflammation		CXCL1		
Fibrosis				CXCL-1
Cell death				
Oxidative stress		ATM		NF-κΒ
DNA damage				HMOX1, PRKDC
alamarBlue and Comet assay toward NC (unexposed) Lab 1				
	PBS 10%		NM-300K exposure	
	Apical	Basolateral	Apical	Basolateral
Cytotoxicity	−	−	+	+
DNA damage	−	−	−	−

**Table 3 nanomaterials-14-01888-t003:** Summary of results for Laboratory 2 (Lab 2). Changes in gene expression (RT^2^ profiling assay) and in cytotoxicity (alamarBlue) and DNA damage (comet assay) 24 h after air-liquid interface exposure with PBS 10% and 20 µg/cm^2^ of NM-300K. Exposure groups are compared to NC from Laboratory 2.

RT^2^ Profiling Assay Lab 2				
NM-300K Exposure Toward PBS 10% Lab 2				
Regulation	Apical		Basolateral	
	Down-	Up-	Down-	Up-
Inflammation	IL-1β, IP-10, RANTES			
Fibrosis	Cd11b1			CXCL-1
Cell death	BCL-2			RIPK1, Caspase9
Oxidative stress				
DNA damage				HMOX1
alamarBlue and Comet assay toward NC (unexposed) Lab 2				
	PBS 10%		NM-300K exposure	
	Apical	Basolateral	Apical	Basolateral
Cytotoxicity	−	−	−	−
DNA damage	−	−	+	+

## Data Availability

Data are available from the researchers on request.

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
