# Peer review of "Toward Standardization of a Lung New Approach Model for Toxicity Testing of Nanomaterials"

_nanomaterials, 2024, doi:10.3390/nano14231888_

Round 1

Reviewer 1 Report

Comments and Suggestions for Authors

the work presented here is interesting, as i tries to standardize experiment via the comparison between two laboratories. The results could be convincings, but some flaws impact the quality of the article.

First, NM-330k, is never described, expect in a the legend of a figure, and at the end of the article. Please precise that it is a silver np as early as possible in introduction and in M&M section. Second, why this nano ?

Third, considering the cell model, could the authors justify it ? EA.hy926 cells line is an hybrid between huvec and A549 cells. Could it be interactions between those cells an dthe A549 at the apical side? 

Fourth, the authors mixed a ratio of 1 A594 / 2 THP1, while in the natural environment, the ratio is 10 epithelial cells/1 macrophage. Why this choice ? 

Fifth, if i understand well, no surfactant was used during ALI experiments. It is very uncommon. How the dessication of cells is avoided ? 

The discrepency in gene expression is very puzzling. The justification is insufficient and alters the all conclusion of the work.

Author Response

Reviewer 1

Comments and Suggestions for Authors

The work presented here is interesting, as tries to standardize experiment via the comparison between two laboratories. The results could be convincing, but some flaws impact the quality of the article.

First, NM-330k, is never described, expect in the legend of a figure, and at the end of the article. Please precise that it is a silver np as early as possible in introduction and in M&M section. Second, why this nano?

1) We agree and have accordingly mentioned in the introduction NM-300K silver nanoparticles (Lines 70- 74) and modified the Materials and Methods section, specifically paragraph 2.2 (Lines 131-139), to include a description of NM-300K and the rationale for its selection as the reference particle in this study.

2) NM-300K engineered spherical silver NMs with a pristine size of less than 20 nm and are classified as nanoparticles, as their size is below 100 nm.

Third, considering the cell model, could the authors justify it? EA.hy926 cells line is an hybrid between huvec and A549 cells. Could it be interactions between those cells and the A549 at the apical side? 

3) Thank you for highlighting this important point. EA.hy926 cells exhibit significant morphological differences compared to A549 cells. The distinct morphology of these cells makes them easily recognizable (Camassa and Elje, 2022). In the experimental procedure, EA.hy926 cells are seeded on the basolateral side of the inserts and placed in the incubator for a minimum of 4 hours. Subsequently, the inserts are inverted within a 6-well plate, and A549 cells are seeded on top. This step is crucial for maintaining the separation of the two cell lines. No migration of EA.hy926 cells is observed on the apical side in the electron microscopy micrographs (Camassa and Elje et al., 2022).

Fourth, the authors mixed a ratio of 1 A594 / 2 THP1, while in the natural environment, the ratio is 10 epithelial cells/1 macrophage. Why this choice? 

We appreciate your comment and thank you for bringing this to our attention. We accordingly modified and add additional information in the paragraph 2.1 (Material and Methods) lines 108-116. We observed that the actual ratio of dTHP-1 to A549 was significantly lower and more closely aligned with the in vivo situation.

Fifth, if i understand well, no surfactant was used during ALI experiments. It is very uncommon. How is the desiccation of cells avoided? 

Thank you for bringing this to our attention. We appreciate your comment and agree that A549 cells may encounter challenges in surfactant production, potentially related to the specific cell batch used or cell passage. We accordingly modified and added additional information in a limitation paragraph at the end of the discussion (lines 617 - 624). Even though we didn’t observe any dryness, we plan to further optimize the model, and avoid possible dryness effect, to include the incorporation of commercially available surfactant.

The discrepancy in gene expression is very puzzling. The justification is insufficient and alters the whole conclusion of the work.

We appreciate your comment and thank you for bringing this to our attention. Laboratory comparisons are challenging and depend on various factors that can influence the study's outcome. Although we employed identical protocols for cell culture, we recognize that it is essential for laboratory comparisons to utilize the same batch of medium, serum, and supplements. We also acknowledge some limitations in the models. Accordingly, we have added an additional explanation, including a limitations paragraph at the end of the discussion (see lines 610-631). Furthermore, we provided more results on negative controls between laboratories (Paragraph 3.3.1, figure 5 and tables 2 and 3) to help clarify this gene expression discrepancy. We also add group gene names in the figure 3 and 4. We acknowledge this issue, and in the future, we will implement new guidelines to better standardize the model for assessing the toxicity of nanomaterials. Nevertheless, we believe that this study is important for enhancing knowledge and developing new methodological approaches.

Reviewer 2 Report

Comments and Suggestions for Authors

In my opinion, this is a valuable contribution to the standardization of NAMs in toxicological tests. Above all, the performance in different laboratories and thus the verification of the reproducibility of results is often neglected and is an important component in the verification of the applicability of the methods.

In my opinion, what is missing is a description of the uptake of the silver particles by the cells. Especially with regard to the described inter-laboratory differences, this would be an important indication of the divergent results of the respective laboratories. In the case of silver particles, microscopic observations could be made. Cell-internal agglomerates can be observed in bright field and single particle occurrences could be visualized with e.g. dark field microscopy. Quantification of the silver content using mass spectroscopy methods would be recommended. This would then also involve a more precise characterization/description of the particle deposits in the cells (agglomerates/individual particles, size).

Which cell type is primarily involved in the uptake of particles? In the lungs, it is known that alveolar macrophages in particular are involved in the uptake and clearance of inhaled particles. Is there any information that this is also the case in this model? A549 cells, an epithelial cell line, also take up particles to a considerable extent, which is not the case for this cell type in the lung. Although this cell line is widely used, I think it behaves atypically with regard to particle uptake. Since this in vitro model is so complex and elaborate, it would all the more important to compare and correlate the particle uptake and resulting findings with existing in vivo results from inhalation studies/instillation studies (mouse/rat). 

In the introduction, reference is made to the 3Rs principle; however, it should also be borne in mind that FCS must be added to the cell culture medium for the cultivation of all cell lines used. The production and use of FCS should not be viewed uncritically.

With regard to the THP-1 cells, it should be described how the cells were differentiated (duration and concentration of PMA); there is a lack of information on whether the differentiation process was identical in the respective laboratories.

The NANOGENOTOX protocol was used for the dispersion of NM300K. It is important to note here that BSA is used, which can change the particle properties in a not insignificant way. One idea on my part would be to use this dispersion protocol without BSA, which should not cause any problems in the case of silver particles. In any case, the control for the ALI exposure should not only consist of 10% PBS, but a vehicle control (with BSA) must also be included. In addition, the decrease in cell viability with PBS administration is a cause for concern. It should be ensured that the cells do not exhibit any drying artifacts during the ALI exposure period.

Is there an explanation/significance of the cell numbers used in the model? Why are twice as many differentiated THP-1 cells used as A549 cells, does this have a justifiable physiological background?

 Overall, I think the reader understands why NAM and a standardization of these methods are necessary.

However, due to the divergent results, greater importance should be attached to the observation/description of the cell uptake of particles. An in vitro/in vivo correlation of the results is unfortunately missing and a quantification in the sense of a dose-effect description would also be important here.

Author Response

Which cell type is primarily involved in the uptake of particles? In the lungs, it is known that alveolar macrophages in particular are involved in the uptake and clearance of inhaled particles. Is there any information that this is also the case in this model? A549 cells, an epithelial cell line, also take up particles to a considerable extent, which is not the case for this cell type in the lung. Although this cell line is widely used, I think it behaves atypically with regard to particle uptake. Since this in vitro model is so complex and elaborate, it would all the more important to compare and correlate the particle uptake and resulting findings with existing in vivo results from inhalation studies/instillation studies (mouse/rat). 

We appreciate your comment. In this context, I would like to emphasize that in our previous paper and the initial section of the 3D lung alveolar model standardization, we employed dark field microscopy and inductively coupled plasma mass spectrometry (ICP-MS) to measure silver content, as well as electron microscopy to investigate cellular uptake. Notably, in our 3D lung alveolar model, both dTHP1 cells and, to a lesser extent, A549 cells are capable of engulfing NM-300K nanoparticles (Camassa and Elje, 2022). We agree with your observation and have added an additional paragraph at the end of the discussion (lines 623-629) to clarify this point. It would be beneficial to include electron microscopic images to examine the differences in nanoparticle uptake and how this may influence the results. In the future, we will include a comparison with in vivo studies to validate the model. Thank you for your insightful comment; we will incorporate this feedback into our future studies on model standardization.

In the introduction, reference is made to the 3Rs principle; however, it should also be borne in mind that FCS must be added to the cell culture medium for the cultivation of all cell lines used. The production and use of FCS should not be viewed uncritically.

We also acknowledge the significance of the comment regarding serum. We recognize that the use of fetal bovine serum (FBS) does not render our model entirely animal-free. We understand the necessity of utilizing various types of cells that may not require this type of supplementation for growth. Unfortunately, many of these cell culture models still lack validation, and limitations such as availability and budget constraints are influencing our choice of cell models. We agree that we should not uncritically accept the use of FBS, and we will implement additional guidelines in future experiments to align our model more closely with animal-free practices and adhere to the principles of the 3Rs.

With regard to the THP-1 cells, it should be described how the cells were differentiated (duration and concentration of PMA); there is a lack of information on whether the differentiation process was identical in the respective laboratories.

We appreciate the feedback and have added additional information in paragraph 2.1 (lines 108 to 116) to clarify the text.

The NANOGENOTOX protocol was used for the dispersion of NM300K. It is important to note here that BSA is used, which can change the particle properties in a not insignificant way. One idea on my part would be to use this dispersion protocol without BSA, which should not cause any problems in the case of silver particles. In any case, the control for the ALI exposure should not only consist of 10% PBS, but a vehicle control (with BSA) must also be included. In addition, the decrease in cell viability with PBS administration is a cause for concern. It should be ensured that the cells do not exhibit any drying artifacts during the ALI exposure period.

We appreciate your comment and thank you for bringing this to our attention. In NanoGenoTox, the dispersion vehicle consists of 0.05% BSA in MQ water. Nebulizing a solution containing 0.05% BSA can create bubbles in the nebulizer, which may affect cell viability results when compared to the unexposed control. However, we did not observe any dry artifacts, as A549 cells have also been shown to produce surfactant. We found that more than 80% of the cells remained viable after nebulization of our vehicle, compared to the unexposed control. Tables with numerical data are included in the supplementary material S3.2. Additionally, we have provided further information in the text §2.2, lines 140-145, to clarify this point.

Is there an explanation/significance of the cell numbers used in the model? Why are twice as many differentiated THP-1 cells used as A549 cells, does this have a justifiable physiological background?

Thank you for pointing this out. We value your comment. We accordingly modified and add additional information in the paragraph 2.1 (Material and Methods) lines 108-116. We observed that the actual ratio of dTHP-1 to A549 was significantly lower and more closely aligned with the in vivo situation.

Overall, I think the reader understands why NAM and a standardization of these methods are necessary.

However, due to the divergent results, greater importance should be attached to the observation/description of the cell uptake of particles. An in vitro/in vivo correlation of the results is unfortunately missing and a quantification in the sense of a dose-effect description would also be important here.

We appreciate your comments and feedback, and we accordingly have made changes in the articles. We provided more results on negative controls between laboratories to help clarify the gene expression discrepancy between controls. We added a new paragraph 3.3.1 and tables (Table 2 and 3) for greater clarity. We add also a limitation paragraph at the end of the discussion (see lines 610-631). We acknowledge this issue, and in the future, we will implement new guidelines to better standardize the model for assessing the toxicity of nanomaterials.

Reviewer 3 Report

Comments and Suggestions for Authors

In this manuscript the authors to investigated the possibility of standardization of a lung new approach in vitro model for toxicity testing of nanomaterials.

The manuscript is well written for originality and novelty including the description of the methods employed, the results and discussion.

Minor issue: in introduction section, I suggest to authors the shift the final part  in discussion section (see pag 2, line 73-88) for improve scientifically sound. In this final part of introduction section, I suggest to say something more about the purpose of this study and the approach used for finalize the study.

Author Response

Minor issue: in introduction section, I suggest to authors the shift the final part in discussion section (see pag 2, line 73-88) for improve scientifically sound. In this final part of introduction section, I suggest saying something more about the purpose of this study and the approach used for finalizing the study.

We appreciate your feedback and have made further revisions to the article, specifically at the end of the introduction, in the discussion section and in the conclusion. We added also name of the group’s gene in the Figure 3 and 4, add a new paragraph 3.3.1, and a new Figure 5. We hope we enhance scientific rigor of the paper and clarify its primary objective.

Round 2

Reviewer 1 Report

Comments and Suggestions for Authors

I am satisfied of the author's answer, despite, I am doubtful on the justification about the cell ratio, I think that it's a strong weakness of this study.

Author Response

I am satisfied of the author's answer, despite, I am doubtful on the justification about the cell ratio, I think that it's a strong weakness of this study.

Thank you for bringing this to our attention. We appreciate your comment and have added more information in several sections of the article. We hope this will clarify our decision to continue using the dTHP1 cell density in relation to the A549 epithelial cell line. In Section 2.1, line 122, we reference entirely new data and tables included in the supplementary material (Tables S4 and S6, and Supplementary Figure 1). Additionally, we have added a new paragraph in Section 2.2, the Materials and Methods (immunofluorescence). In the limitations paragraph at the end of the article, we have included a statement regarding the challenges associated with the adherence and damage of dTHP1 (Section 4, lines 632-639).